# Role of Kindlin-2-Expressing Extracellular Vesicles in the Invasiveness of Triple Negative Breast Cancer Tumor Cells

**DOI:** 10.3390/cells14131034

**Published:** 2025-07-07

**Authors:** Neelum Aziz Yousafzai, Mark F. Santos, Yeaji Kim, Nofar Avihen Schahaf, Kim Zielke, Lucia Languino, Khalid Sossey-Alaoui, Aurelio Lorico

**Affiliations:** 1School of Medicine, The MetroHealth System, Case Western Reserve University, Cleveland, OH 44106, USA; nyousafzai@mgh.harvard.edu (N.A.Y.); yxk811@case.edu (Y.K.); 2College of Medicine, Touro University Nevada, Henderson, NV 89014, USA; msantos17@touro.edu (M.F.S.); nschahaf@touro.edu (N.A.S.); kzielke@student.touro.edu (K.Z.); 3Department of Pharmacology, Physiology, and Cancer Biology, Thomas Jefferson University, Philadelphia, PA 19107, USA; lucia.languino@jefferson.edu

**Keywords:** breast cancer, metastasis, extracellular vesicles, exosomes, kindlin, nucleus, nuclear transport

## Abstract

Metastatic breast cancer (BC) is a major cause of cancer-related deaths among women. Its progression is influenced by extracellular vesicles (EVs) released by BC cells, which modulate distant tissue environments to promote metastasis. We previously identified the oncogenic protein Kindlin-2 (K2) as a key driver of BC metastasis, including its role in the nucleus in regulating cell senescence. Here, we investigated whether K2-containing EVs facilitate both autologous (cancer-to-cancer) and heterologous (cancer-to-stroma) communication to promote metastasis. We found that 10–15% of EVs from metastatic BC cells contained K2, while this subpopulation was nearly absent in the EVs from K2-knockout (KO) cells, indicating selective packaging. These EVs transferred K2 to recipient K2-KO cells, where they accumulated in the nucleus. Using a 3D tumorsphere assay, we showed that K2+ EVs enhanced cancer cell invasiveness. Moreover, K2+ EVs activated fibroblasts into a cancer-associated phenotype, increasing α-SMA and FAP expression. Conditioned media from these activated fibroblasts further boosted cancer cell invasion. These results show that EV-associated K2 is actively transferred to recipient cells and regulates metastasis through nuclear signaling, suggesting K2+ EVs are critical mediators of BC progression and potential targets for therapy.

## 1. Introduction

Breast cancer (BC) remains the second most common cause of cancer-related mortality among women in the United States, accounting for over 44,000 deaths each year [1,2,3]. Although many patients are diagnosed at an early, noninvasive stage, approximately 30% will eventually develop metastatic disease—a transition that dramatically worsens prognoses and limits therapeutic options [1,2,3]. In fact, metastasis accounts for nearly 90% of cancer-related deaths. The clinical challenge is further compounded by the genetic heterogeneity of BC, which encompasses at least five distinct molecular subtypes [4,5,6]. Among them, triple negative breast cancers (TNBCs) are especially aggressive, characterized by a high rate of recurrence and a strong propensity for distant metastasis. Strengthening our ability to prevent tumor progression and metastasis would significantly improve the clinical course of these patients.

Over the past decade, extracellular vesicles (EVs) have gained recognition as pivotal vehicles of intercellular communication, influencing a wide range of biological functions in both health and disease. EVs are nano-sized units mainly comprising exosomes (released in the extracellular space by multi-vesicular bodies upon fusion with the plasma membrane), ectosomes (directly shed from the plasma membrane), large vesicles, and apoptotic bodies [7]. Present in all body fluids at very high concentrations, ranging from hundreds of millions to billions per mL [8], they contain nucleic acid and protein cargoes protected by a phospholipid membrane. While trillions of EVs circulate in the blood of each individual, cancer patients have even higher levels and, most importantly, their EVs carry additional cancer-specific cargo [8,9,10,11]. In the context of BC, EVs play multifaceted roles by enhancing tumor cell invasiveness, promoting motility, and altering the surrounding stromal environment. Beyond these local effects, EVs also contribute to systemic changes that facilitate metastatic spread [9,10,12,13]. They can transport oncogenic proteins [14,15,16], lipids, and various nucleic acids to a range of recipient cells, including neighboring BC cells [17], cancer-associated fibroblasts (CAFs), macrophages [18,19], and endothelial cells [20]. Also, EVs secreted by CAFs have been shown to drive cancer progression by transferring bioactive cargo back to tumor cells [21].

Since EVs are involved in the pathogenesis and/or in the progression of BC, the correct therapeutic approach would be to impair their cellular entry or delivery of their cargo. Currently, it is not clear how a minute cargo contained in nanometer-sized entities results in a biological effect without concentrating EVs in a specific sub-cellular compartment, specifically the nucleus.

We previously discovered a novel intracellular pathway that allows EVs to transfer their cargo into the nuclear compartment of recipient cells, where they alter gene expression. Upon entry into recipient target cells by endocytosis, EVs are carried by late endosomes, defined by Rab7 GTPase, into type II nuclear envelope invaginations (NEIs), composed of both inner and outer nuclear membranes. Their entry into NEIs and docking to the outer nuclear membrane are mediated by the VOR complex, made of three proteins: the vesicle-associated membrane protein-associated protein A (VAP-A) on the outer nuclear membrane, the oxysterol-binding protein (OSBP)-related protein-3 (ORP3), and the Rab7 on late endosomes [22,23,24]. EV cargoes are then transferred to the nucleoplasm via nuclear pores.

Kindlins are a small gene family (with three members) of FERM domain-containing adaptor proteins that are crucial for integrin activation. The dysregulated expression or function of Kindlins has been implicated in multiple human diseases, including BC [25,26,27]. Kindlin-2 (K2), the most broadly expressed isoform, is essential for normal development; a complete loss of K2 in mice results in embryonic lethality, whereas heterozygous mice appear phenotypically normal under baseline conditions but display impairment in angiogenesis, hemostasis, and cytoskeletal organization when subjected to physiological stress [28,29,30].

We previously found that the EVs secreted by PC3 prostate cancer contained K2 and were able to regulate the αVβ3-mediated activation of prostate cancer progression and metastasis [31]. Given that we have shown that K2 is enriched in cancer cell-secreted EVs [31], and that K2 can also be found in the nucleus where it regulates cancer cell senescence [32], we hypothesized that K2-containing EVs deliver K2 and its associated cargo to stromal as well as cancer cells, in both an autologous and a heterologous mode, and play a major role in the activation of the metastatic process.

In BC, we previously identified that K2 can be found not only in the cytoplasm, but also in the nucleus, where it regulates cancer cell senescence. Here, we hypothesized that K2-containing EVs deliver K2 and its associated cargo to the nucleus of other cancer cells (autologous communication) and to stromal/immune cells (heterologous communication) and play a major role in the activation of the metastatic process.

## 2. Methods

### 2.1. Cell Culture

The human MDA-MB-231 (231) and mouse 4T1 breast cancer cell lines were acquired from ATCC and cultured in RPMI-1640 medium (#10-041-CV, Corning Inc., New York, NY, USA) supplemented with 10% fetal bovine serum (FBS; #26140079, Thermo Fisher Scientific, Waltham, MA, USA), 2 mM L-glutamine (#25030081), and antibiotics (100 U/mL penicillin and 100 μg/mL streptomycin; #15140122). The cells were maintained at 37 °C in a humidified atmosphere containing 5% CO_2_. To generate K2-knockout (KO) lines, the cells were electroporated with a ribonucleoprotein complex composed of Cas9 protein and three pooled sgRNAs targeting either the human or mouse K2 gene (Synthego, Redwood, CA, USA), following the manufacturer’s protocol and established procedures [33]. Scrambled sgRNAs were used as non-targeting controls [34]. Knockout efficiency was validated by Western blot, and if less than 80%, a second electroporation was performed.

### 2.2. Isolation and Characterization of Small Extracellular Vesicles (EVs)

EVs were isolated from a conditioned medium of parental and K2-knockout (K2-KO) 231 and 4T1 cells. The cells were seeded at a density of 2.5 × 10^5^ cells per well in 6-well plates coated with 20 μg/mL poly(2-hydroxyethyl methacrylate) (poly-HEMA, #P3932, Sigma-Aldrich, Burlington, MA, USA) to inhibit adherence, following the method described by Rappa et al. [35] and maintained for 72 h in serum-free RPMI-1640 supplemented with a 2% B-27 supplement (#17504044, Thermo Fisher Scientific). At the time of EV harvest, cell viability exceeded 90% as determined by trypan blue exclusion, and the average harvest density was approximately 3.5 × 10^5^ cells per well.

The conditioned media were subjected to differential centrifugation: 300× *g* for 10 min and 1200× *g* for 20 min to remove cells and debris, followed by 10,000× *g* for 30 min at 4 °C. The resulting supernatant was centrifuged at 100,000× *g* for 60 min at 4 °C using a Beckman Coulter Optima ultracentrifuge with an SW41Ti rotor. All centrifugation steps were performed using maximum acceleration and deceleration (brake) settings. The EV pellet was resuspended in 200 μL of sterile PBS and stored at −80 °C until further use.

The size and concentration of EVs were estimated using nanoparticle tracking analysis (NTA) on the ZetaView platform (Particle Metrix GmbH, Holly Springs, NC, USA). Data acquisition was performed in scatter mode using a 488 nm laser, capturing videos at 30 frames per second across 11 positions along the *z*-axis, with each video lasting 2 s per position. The instrument was set to a camera gain of 10 and a trace length of 15. Electron microscopy (EM) was not performed in this study, as our primary focus was on EV-associated functional and molecular characteristics. We acknowledge that NTA does not provide morphological validation, and this is noted as a limitation.

### 2.3. Immunocytochemistry

K2-deficient 231 or 4T1 cells were plated at a density of 50,000 cells per well in 8-well #1.5H glass-bottom μ-slides (#80827, Ibidi, Fitchburg, WI, USA). Once the cells adhered, 1 × 10^9^ EVs from the corresponding parental cell line were applied and incubated for 5 h at 37 °C. Following incubation, the cells were rinsed with PBS, fixed using 4% paraformaldehyde for 15 min, permeabilized in 0.2% Tween 20/PBS for another 15 min, and then blocked in 1% bovine serum albumin (BSA; #001-000-162, Jackson ImmunoResearch, West Grove, PA, USA) for 1 h at room temperature (RT).

The samples were incubated with primary antibodies against VAP-A (#H00009218-M01, clone 4C12, Abnova, Taipei, China, 1:50) and Kindlin-2 (#PA5-59200, Invitrogen, Waltham, MA, USA 1:100) for 1 h at RT, followed by washing and incubation with AlexaFlour647-conjugated anti-mouse (#A21237, Invitrogen, 1:1000) or AlexaFlour488-conjugated anti-rabbit (#A11070, Invitrogen, 1:1000) secondary antibodies for 30 min. All antibodies were diluted in a permeabilization buffer containing 1% BSA. Imaging was performed in PBS using the Nanoimager super-resolution microscope (Oxford Nanoimaging (ONI), San Diego, CA, USA) fitted with a 100× oil-immersion objective. Fluorescence signal intensity was analyzed using Fiji (ImageJ) software, version 21.0.7.

### 2.4. Direct Stochastic Optical Reconstruction Microscopy (D-STORM)

EVs were immobilized on microfluidic chips provided in the EV Profiler Kit (#EV-MAN-1.0, ONI) and immunolabeled for surface or intravesicular markers, following the manufacturer’s protocol. EVs were fixed using F1 fixation solution (ONI) for 10 min at RT. For permeabilization, 0.01% Triton X-100 in PBS was applied for 10 min at RT. Immunolabeling was carried out using fluorescently conjugated antibodies against K2 (#MAB2617, EMD Millipore, Burlington, MA, USA) and pan-tetraspanin markers (CD9, CD63, CD81; provided in the kit). The antibodies were diluted in a permeabilization buffer containing 1% BSA and 0.01% Triton X-100. Labeled EVs were incubated for 50 min at RT, followed by a second fixation step with an F1 solution for 10 min. The chips were washed in PBS and imaged immediately in a freshly prepared d-STORM imaging buffer (ONI).

Single-molecule fluorescence data consisting of 2000 frames per channel was sequentially acquired using the Nanoimager S Mark II system (ONI). Laser power was set to 45% and 50% for the 640 nm and 488 nm lasers, respectively. Imaging was performed using an Olympus 100× 1.4 NA oil immersion super apochromatic objective with the angle of illumination set to 52.5°. Multichannel registration was calibrated at the start of each imaging session using 0.1 μm TetraSpeck beads (#T7279, Thermo Fisher Scientific) to account for chromatic aberration. Images were acquired under highly inclined illumination conditions to minimize background.

Raw data was processed using NimOS software (version 1.19.15, ONI), and downstream analysis was conducted using CODI (https://alto.codi.bio/, accessed on 1 July 2025)), ONI’s cloud-based localization microscopy analysis platform. The analysis included localization filtering, drift correction, and hierarchical density-based clustering to identify single EVs and assess marker colocalization. A minimum of 3500 individual EVs were analyzed per sample.

### 2.5. Western Blot

Cell lysates were generated using an ice-cold lysis buffer containing 1% Triton X-100, 150 mM NaCl, and 50 mM Tris-HCl (pH 8.0) supplemented with a Halt™ Protease Inhibitor Cocktail (#78425, Thermo Fisher Scientific). The samples were incubated on ice for 30 min and centrifuged at 10,000× *g* for 5 min at 4 °C. The resulting supernatants were combined with SDS sample buffer (#1610747, Bio-Rad, Hercules, CA, USA) and heat-denatured at 95 °C for 5 min. The EV samples were lysed directly in SDS sample buffer and processed identically.

Protein samples were separated on 4–20% Tris-glycine precast gels (#4561095, Bio-Rad) and transferred to nitrocellulose membranes (#88018, Thermo Fisher Scientific). The membranes were blocked for 1 h at RT in PBS containing 1% BSA, then incubated overnight at 4 °C with primary antibodies targeting human FAP (#66562, Cell Signaling, Danvers, MA, USA 1:1000), mouse FAP (#EWI042, Kerafast, Boston, MA, USA, 1:1000), Kindlin-2 (#MAB2617, EMD Millipore, 1:2000), β-Actin (#A2228, Sigma-Aldrich, 1:5000), calnexin (#MA3-027, clone AF18, Invitrogen, 1:500), CD9 (#SC-20048, clone P1/33/2, Santa Cruz Biotechnology, Dallas, TX, USA, 1:500), CD81 (#MA5-13548, clone 1.3.3.22, Invitrogen, 1:500), or α-SMA (#19245S, Cell Signaling, 1:1000). After washing, membranes were incubated for 30 min at RT with AlexaFluor488-conjugated secondary antibodies (#A11017, Invitrogen, 1:2000). Signals were detected and quantified using the iBright FL1000 imaging system (Thermo Fisher Scientific).

### 2.6. Tumorsphere Invasion

The EVs derived from TNBC cell lines were used to investigate the effect on tumorsphere invasion in a 3-dimensional model of tumorspheres grown in Matrigel as previously described [36]. Briefly, cells were seeded in 96-well ultralow attachment (ULA) plates at a density of 1.0 × 10^3^ cells per well and centrifuged at 125× *g* for 10 min at RT. On day 3, 90 μL of Matrigel diluted at a 1:1 ratio with complete medium was gently added to each well. Tumorsphere growth and invasion were monitored by imaging every 48 h using a Leica CMi1 microscope, with observations continuing over an 11-day period. EVs were isolated from the cultures of parental and K2-KO derivatives to investigate the autologous effect of EVs. We also used fibroblast-cancer cells co-cultures to investigate the heterologous effect of EVs.

### 2.7. Statistical Analysis

All experiments were conducted in a minimum of three independent replicates. The results are expressed as mean ± standard deviation (SD). Comparisons between two experimental groups were assessed using unpaired, two-tailed Student’s *t*-tests. A *p* value less than 0.05 was considered statistically significant. Data visualization and statistical calculations were performed using GraphPad Prism (version 10).

## 3. Results

To explore the potential role of K2 in EV biology, we first isolated EVs from 231 and 4T1 BC cells using a standard ultracentrifugation-based protocol. To ensure that EVs were produced independently of cell attachment, the cultures were maintained in serum-free media on poly-HEMA-coated plates. Direct stochastic optical reconstruction microscopy (d-STORM) and Western blotting for established EV markers were used to confirm particle identity and purity.

Nanoparticle tracking analysis (NTA) was then performed to assess whether the genetic deletion of K2 affects EV secretion. The size distribution and concentration of EVs from parental 231 and 4T1 cells and their K2-knockout (K2-KO) counterparts were comparable across both cell lines (Figure 1A). For 231 cells, the average EV concentration was 5.06 ± 1.14 × 10^9^ particles/mL for parental cells and 5.13 ± 0.61 × 10^9^ particles/mL for K2-KO cells (*p* = 0.93), with corresponding sizes of 135.0 ± 13.1 nm and 136.1 ± 14.7 nm, respectively (*p* = 0.93). For 4T1 cells, the average EV concentration was 7.52 ± 3.28 × 10^9^ particles/mL for parental cells and 8.92 ± 4.77 × 10^9^ particles/mL for K2-KO cells (*p* = 0.70), with corresponding sizes of 135.7 ± 14.9 nm and 114.3 ± 17.5 nm, respectively (*p* = 0.18). These results indicate no statistically significant difference in overall EV yield or size distribution. The lack of changes suggests that K2 is not required for bulk EV production or release and that K2-KO does not substantially impact EV biogenesis under the conditions tested.

We next asked whether K2 is physically incorporated into EVs. Using d-STORM, we observed a strong co-localization of K2 with pan-tetraspanin markers on the surface of the EVs derived from parental 231 and 4T1 cells (Figure 1B). In contrast, EVs from K2-KO cells showed a marked reduction in K2 signal. Quantification revealed that approximately 15% and 10% of EVs from 231 and 4T1 parental cells, respectively, were K2–positive, while this population was virtually absent in the K2-KO EVs (Figure 1C). This supports the notion that K2 may not be merely passively incorporated but is selectively enriched in a defined EV subpopulation.

We validated these imaging findings by performing a Western blot analysis on both the whole-cell lysates and the purified EV fractions (Figure 1D). K2 was readily detectable in EVs from both the 231 and 4T1 control cells but absent in EVs from the K2-KO cells. The canonical EV markers CD9 and CD81 were present in all the samples, while calnexin, a marker typically excluded from EVs, was undetectable, confirming the purity of EV preparations.

We then investigated whether K2 carried by EVs could reach the nucleus—a behavior previously observed for specific EV cargoes in cancer cells [24,37]. K2-KO cells were incubated with the EVs derived from parental (K2–expressing) cells for 5 h, followed by immunolabeling for K2 and VAP-A, a marker of the outer nuclear membrane. Confocal microscopy revealed the presence of K2 puncta in the nuclei of both the 231 and 4T1 K2-KO cells following EV treatment (Figure 1E), indicating successful delivery and nuclear trafficking of EV-associated K2. In contrast, no K2 signal was observed in the nuclei of the untreated K2-KO cells (no-EV control, Figure 1E), confirming that the observed fluorescence was not due to residual endogenous expression or antibody background but was specifically derived from the added K2^+^-EVs.

Quantification of nuclear K2 puncta per cell showed consistent and robust nuclear accumulation in both cell lines upon treatment with parental EVs (Figure 1F). Each data point represents an individual cell, and the distribution supports consistent nuclear delivery across the cell population.

Employing a 3D invasion assay, we showed that the invasion potential of the tumorspheres derived from K2-KO 231 cells was significantly enhanced in the presence of the conditioned media (CM) of parental cells, compared to the CM of their K2-KO derivatives (Figure 2A). To determine whether this effect was caused by the EVs present in the CM, we isolated EVs from both the parental and K2-KO 231 cells and confirmed that the activation of the invasion potential was indeed driven by the K2^+^-EVs (Figure 2B). These data evidenced an autologous effect of EVs on cancer cells. We then exposed normal mouse 3T3 fibroblasts to CM from mouse 4T1 BC cell cultures and found a significantly induced expression of α-smooth muscle actin (α-SMA) and fibroblast activation protein (FAP), both markers of the myofibroblast-activated phenotype of CAFs, a result that we replicated in human WI-38 fibroblasts treated with CM from 231 BC cells (Figure 2C), therefore supporting the role of BC-secreted factors in the acquisition of the CAF phenotype by normal fibroblasts, i.e., in a heterologous context. We further confirmed that BC-derived EVs are indeed required for fibroblast activation by treating 3T3 or WI-38 fibroblasts with EVs derived from mouse 4T1 or 231 BC cells, respectively (Figure 2D). EV exposure resulted in a significantly induced expression of α-SMA and FAP. Moreover, the K2-KO 231-derived tumorspheres treated with the CM of WI-38 fibroblasts activated by 231-derived EVs (CAFs) exhibited enhanced invasion compared to the tumorspheres derived from treatment with the CM of naïve WI-38 fibroblasts (Figure 2E). These findings were reproduced with mouse 3T3 fibroblasts and 4T1-derived tumorspheres (Figure 2F). Thus, we showed that factors secreted by the BC-educated CAFs have the capability to activate the invasive potential of BC tumorspheres in a paracrine pathway, and that K2 may be an important factor in this process.

## 4. Discussion

The acquisition of metastatic phenotypes is responsible for the death of the majority of cancer patients [1,3]. Metastatic breast carcinoma (BC) is the second leading cause of cancer-related deaths in women, accounting for more than 44,000 deaths and 281,000 new cases every year in the US [1,2,3]. An incomplete understanding of the process of the invasion-metastasis cascade of BC tumors has prevented the development of curative therapeutic strategies. It is now well established that cancer cells exert a systemic effect that modulates the microenvironment in distant organs to regulate the metastatic process [38,39]. Such a systemic effect has been attributed, in part, to the EVs that are secreted by tumor cells and shed in circulation, which, in the context of cancer, the EV cargo that contains proteins, DNA, RNA, cytokines and chemokines, regulates cell migration, invasion, metastasis, and modulation of the tumor microenvironment [9,10,12,13]. In fact, a PubMed search using the keywords ‘EVs or exosomes’ and ‘metastasis’ yielded over 4,100 research articles, with the majority published within the past decade, highlighting the rapidly growing significance of this emerging field in cancer metastasis.

One specific component of the EV cargo is Kindlin-2 (K2), which our group and others have established as a major driver of tumor progression and metastasis in BC and other cancers [25,26,27].

In the present study, we used state-of-the-art EV techniques (d-STORM, NTA and Confocal microscopy) to show that K2 is enriched in EVs secreted by MDA-MB-231 and 4T1 BC cells. This information supported the potential for a key role of the K2 present in the EVs secreted by BCs in the regulation of the invasive properties of BCs. To support this supposition, we used a three-dimensional tumorsphere growth and invasion assay to show that indeed K2 in EVs is required for tumorsphere invasion in both autologous and heterologous settings. Collectively, our results indicate that K2 is not only selectively incorporated into breast cancer-derived EVs but also functionally transferred to recipient cells, where it accumulates in the nucleus. This positions EV-associated K2 as a potential regulator of intercellular communication through nuclear signaling pathways. We previously observed dysregulated K2 expression to be associated primarily with human TNBC cell lines as compared to their luminal (T47D & MCF7) and HER2^+^ (SKBR3) counterparts. Moreover, K2 abundance correlated with the metastatic potential of three TNBC progression series, namely the human MCF10A series [40], the murine 4T1 series [41], and the NME series [42]. Likewise, immunohistochemical staining for K2 expression in human BC tissue specimens showed K2 expression to be significantly dysregulated across all human BC subtypes, especially TNBCs, and across increasing tumor grades, with increased expression compared with normal breast tissue and robust K2 expression correlated with reduced overall survival in BC patients. All those findings suggest that aberrant K2 expression occurs frequently in advanced-stage and metastatic TNBCs. We previously reported that the genetic inactivation of K2 significantly inhibited 3D tumorsphere invasion, impaired TNBC growth and metastasis, relative to their K2-expressing counterparts and resulted in profound transcriptomic alterations in K2-deficient cells, including the gene signatures associated with EMT programs, extracellular matrix remodeling, oncogenic Ras signaling, and vascular development and angiogenesis [33]. Accordingly, we established the importance of dysregulated K2 expression in driving TNBC metastasis by (i) inducing EMT programs via the suppression of miR-200b expression [43]; and (ii) activating a feed-forward signaling loop that promotes the 3D-outgrowth and migration of TNBCs in vitro, doing so by inducing the expression of CSF-1, EGF, and TGF-β1 [44].

Our present findings together with our previously published studies described above support the conclusion that established a systemic role of tumor K2 in hematopoiesis remodeling to regulate the immune evasion of triple-negative breast cancer tumors [34].

## 5. Conclusions

Our findings support the hypothesis that K2 operates as a central regulator of both local invasion and distal pre-metastatic niche formation. Future studies will focus on validating the pro-metastatic role of K2^+^-EVs in vivo. We will specifically investigate whether K2+ EVs suppress anti-tumor immunity by modulating hematopoietic progenitors, dendritic cells, or myeloid-derived suppressor cells (MDSCs) in vivo. We will further dissect the molecular mechanisms underlying the nuclear translocation of K2^+^-EVs and identify the downstream gene regulatory networks modulated by nuclear K2. Transcriptomic and epigenomic profiling of recipient cells treated with K2^+^-EVs will provide mechanistic insight. Finally, we will develop therapeutic strategies targeting K2^+^-EVs, including antibody-mediated neutralization, uptake inhibition, or the genetic depletion of K2 in tumor cells. Such approaches could serve as adjuncts to current therapies, particularly in patients with therapy-resistant TNBC.

In summary, our data position K2^+^-EVs as critical mediators of tumor invasion, intercellular communication, and immune modulation. Continued investigation into their biological functions and therapeutic vulnerabilities may open new avenues for the treatment of aggressive breast cancers.

## Figures and Tables

**Figure 1 cells-14-01034-f001:**
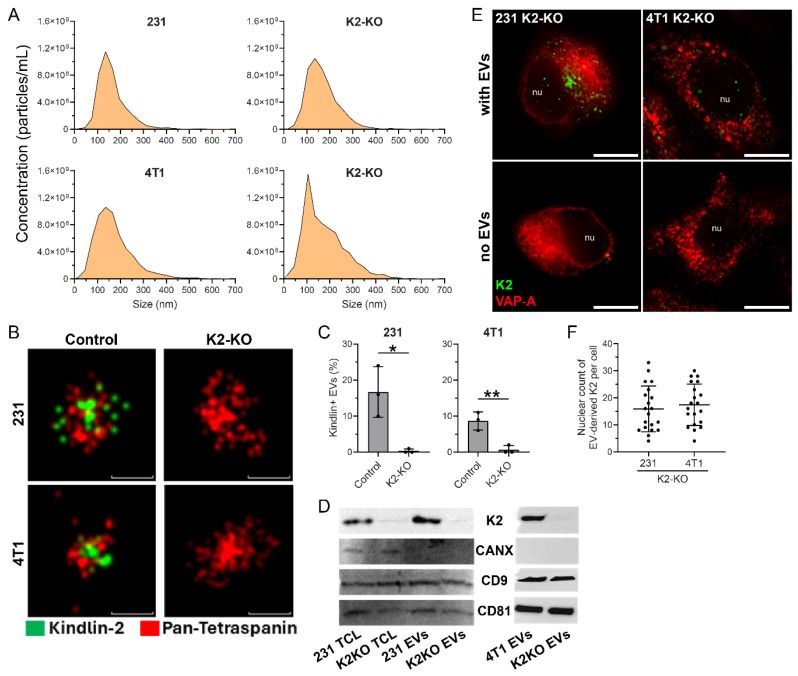
K2 is selectively packaged into breast cancer cell-derived small EVs and transferred to the nucleus of recipient cells. (**A**) Nanoparticle tracking analysis (NTA) showing size distribution and concentration of EVs derived from control and K2-knockout (K2-KO) 231 and 4T1 cells. Both size and concentration profiles were similar between control and K2-KO EVs for each cell line. (**B**) Super-resolution d-STORM images of EVs labeled with pan-tetraspanin (red) and K2 (green). Representative images of control and K2-KO EVs from 231 and 4T1 cells are shown. Scale bars: 100 nm. (**C**) Quantification of K2–positive EVs as a percentage of total EVs in each group. Data represent mean ± SD from at least three independent experiments (*, *p* < 0.05; **, *p* < 0.01). (**D**) Western blot analysis of total cell lysates (TCLs) and EV fractions from control and K2-KO cells. EVs were probed for K2 (K2), calnexin (CANX, negative EV marker), and tetraspanins CD9 and CD81 (EV markers). For uncropped blots, see Appendix A below. (**E**) Confocal microscopy images showing nuclear localization of EV-transferred K2 (green) in K2-KO recipient cells, with co-staining for the nuclear envelope protein VAP-A (red). Images represent 231 and 4T1 K2-KO cells with or without treatment with respective parental EVs. Nu, nucleus. Scale bars: 10 µm. (**F**) Quantification of nuclear K2 puncta per cell in 231 and 4T1 K2-KO recipient cells following incubation with parental EVs. Each dot represents a single cell (*n* = 20), with mean ± SD plotted.

**Figure 2 cells-14-01034-f002:**
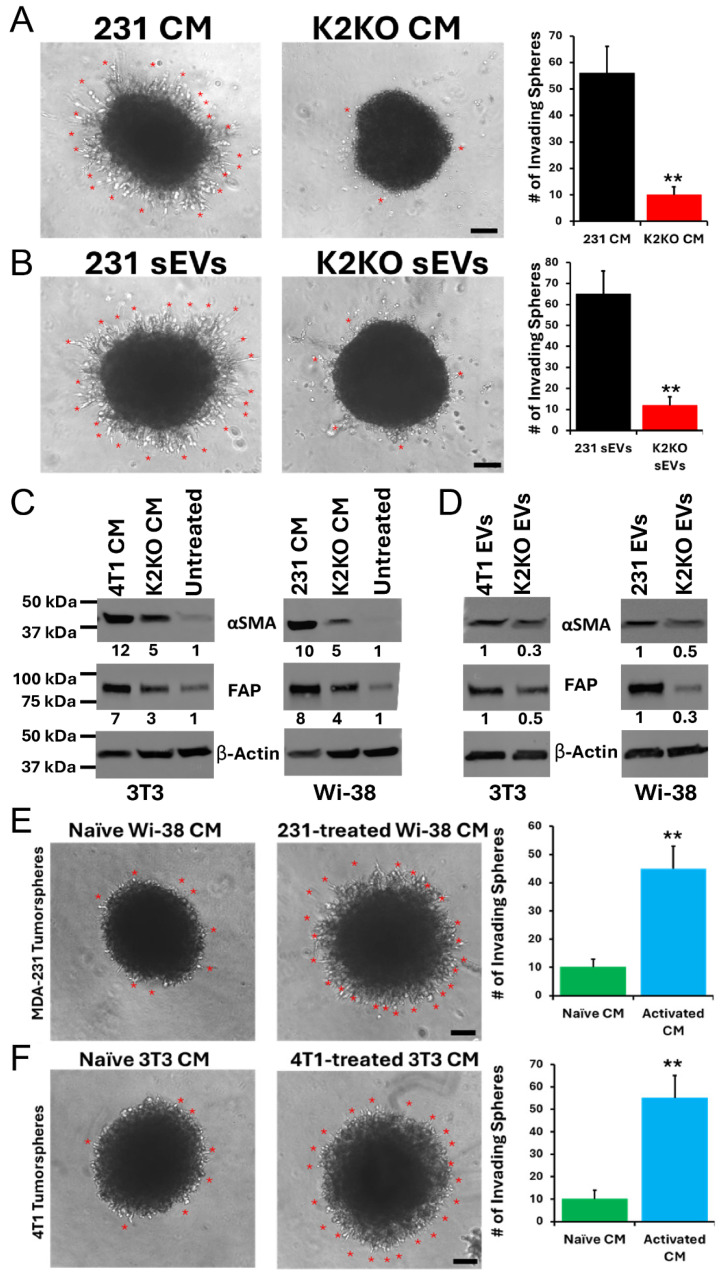
Autologous and heterologous effects of EV-associated K2 on tumor invasion and fibroblast activation. (**A**,**B**) Three-dimensional tumor spheroid invasion assays using 231 cells in the presence of either conditioned media (CM) (**A**) or purified small extracellular vesicles (EVs) (**B**) from parental or K2-knockout (K2-KO) 231 cells. Red asterisks indicate sites of invasion into the surrounding Matrigel. Quantification (right panels) shows a significant reduction in invasive potential with K2-KO CM or EVs. **, *p* < 0.01. Scale bars, 100 µm. (**C**,**D**) Western blot analysis of α-smooth muscle actin (αSMA) and fibroblast activation protein (FAP) expression in mouse 3T3 or human WI-38 fibroblasts following treatment with CM (**C**) or EVs (**D**) from parental or K2-KO 4T1 or 231 cells. Densitometric values normalized to β-actin are shown below each αSMA and FAP band, indicating robust fibroblast activation by parental CM/EVs and markedly reduced activation by K2-KO-derived counterparts. For uncropped blots, see Appendix A below. (**E**,**F**) Functional consequences of fibroblast education by EVs on tumor invasiveness. CM from naïve or EV-educated fibroblasts were applied to 231 (**E**) or 4T1 (**F**) tumor spheroids in 3D invasion assays. Fibroblast education was performed by pre-treatment with EVs from 231 (**E**) or 4T1 (**F**) parental cells. Red asterisks mark invasive protrusions. Quantification (right panels) demonstrates enhanced invasion in the presence of CM from EV-educated fibroblasts. **, *p* < 0.01. Scale bars, 100 µm.

## Data Availability

The data presented in this study are included in the article/Appendix A. Further inquiries can be directed to the corresponding authors.

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
