# Peer review of "Role of Kindlin-2-Expressing Extracellular Vesicles in the Invasiveness of Triple Negative Breast Cancer Tumor Cells"

_cells, 2025, doi:10.3390/cells14131034_

Round 1
Reviewer 1 Report
Comments and Suggestions for Authors
In this paper, Yousafzai and collaborators describe how Kindlin-2-containing EVs transfer the protein horizontally to other tumoral cells in a metastatic breast cancer setting. The subject is quite interesting and promising, but I suggest the authors revise the paper thoroughly and improve the discussion section, which was severely lacking in explaining the topics approached in the abstract and introduction. The paper is also lacking in references throughout the introduction and discussion, which should be remedied. In particular, here are some topics to be considered for revision:
- The abstract should contain max 200 words, as per journal guidelines. The current version has over 400 words. Please correct to comply to the journal policy.
- Introduction: "sEVs are nano-sized units mainly comprised of exosomes (released in the extracellular space by multi-vesicular bodies upon fusion with the plasma membrane), ectosomes (directly shed from the plasma membrane), large vesicles, and apoptotic bodies". This sentence is incorrect and needs to be corrected. If the authors are dividing EVs based on size, ectosomes, large EVs and apoptotic bodies do not comprise small EVs. The reference cited (MISEV2023) also says that: "For example, terms such as ‘small’ and ‘large’ have been commonly used to denote EV populations over the last few years, usually after presumed size-based populations of EVs have been separated with methods such as filtration or differential ultracentrifugation (differential UC, dUC). However, although ‘small’ might generally refer to EVs <200 nm in diameter, there is no strict consensus on upper and lower size cut-offs, and it has also become clear that many separation methods, such as dUC, yield EV populations with overlapping size profiles".
- We suggest the authors thoroughly revise the introduction to add the appropriate references to paragraphs (see page 3-4), such as: "While trillions of sEVs circulate in the blood of each individual, cancer patients have even higher levels and, most importantly, their sEVs carry additional cancer-specific cargo."
- The paragraph starting with "Kindlins are a small gene family [...]" should best be transferred to the discussion section.
- The authors have shown a beautiful invasive model using a 3D approach, but the conclusions were drawn from a single type of invasion assay. Also, the values for # of invading spheres are displayed in dynamite plots, which are not recommended since they do not show data information, such as dispersion and number of replicates. If possible, could the authors present a different assay to confirm these results?
Reviewer 2 Report
Comments and Suggestions for Authors
Article Title :
Role of Kindlin-2-Expressing Small Extracellular Vesicles in the Invasiveness of Triple Negative Breast Cancer Tumor Cells
Summary of the Study :
The authors investigate the role of Kindlin-2 (K2), a FERM domain-containing adaptor protein, in small extracellular vesicles (sEVs) and its involvement in the invasiveness of triple-negative breast cancer (TNBC) cells. Kindlin expression is linked to various pathologies including angiogenesis and hemostasis.
This article explores how K2 is incorporated into sEVs produced by TNBC cells and promotes tumor cell invasion. The authors demonstrate both autologous effects (between tumor cells) and heterologous effects (toward stromal fibroblasts) through K2-mediated nuclear signaling.
They show that sEV-associated K2 can transfer to recipient cancer cells lacking endogenous K2, accumulating in the nucleus, implying a role in transcriptional or chromatin-associated regulation. This nuclear delivery relies on a previously described endosomal–nuclear trafficking pathway, highlighting spatially specific functions of sEV cargo.
Functionally, K2-sEVs enhance the invasiveness of cancer spheroids (autologous effect) and induce a myofibroblastic, cancer-associated phenotype in normal fibroblasts (heterologous effect), as evidenced by increased α-SMA and FAP expression. These “educated” fibroblasts further promote tumor invasiveness, suggesting a feed-forward loop that reinforces the pro-metastatic microenvironment.
Together, the findings propose a novel mechanism by which K2-loaded sEVs orchestrate intra- and intercellular signaling critical for TNBC progression, emphasizing nuclear-targeted EV cargo in metastasis and identifying K2 as a potential therapeutic target to disrupt tumor–stroma crosstalk in aggressive breast cancers.
General Recommendations :
The manuscript is very well written, demonstrating strong command of scientific English. Minor points could be improved but do not compromise clarity or credibility. A light editorial review before final submission is sufficient.
Given the study focuses on EVs, the authors should follow ISEV and MISEV guidelines in their experimental procedures and cite these key references in the introduction.
Some references should be updated, especially recent publications in the rapidly expanding EV field.
Minor Revisions :
- The authors state sEVs are 30–150 nm in diameter. According to the latest MISEV guidelines (doi:10.1002/jev2.12404), sEVs are EVs smaller than 200 nm. Please revise the size description accordingly.
- The authors state that, in a previous study, they discovered a novel intracellular pathway allowing small extracellular vesicles (sEVs) to deliver their cargo into the nuclear compartment of recipient cells (Lines 78–79), and that sEVs enter target cells via endocytosis (references 21–23 of the article). However, the cited article (doi:10.1002/jev2.12132) describes the internalization of EVs by endocytosis, but does not specifically refer to sEVs. This generalization is a shortcut that may cause confusion.
Moreover, how can the authors be certain that all sEVs are internalized exclusively via endocytosis and not through alternative mechanisms, such as direct fusion with the plasma membrane or interactions with other intracellular compartments—with or without membrane fusion? If the authors claim that sEVs are taken up via endocytosis, this assertion should be supported by experimental evidence or by referencing literature that specifically addresses sEV internalization mechanisms.
- The last two paragraphs of the introduction appear redundant; combining them into a single paragraph would improve clarity and flow.
Major Revisions
Materials and Methods :
- K2-deficient cells were generated using CRISPR-Cas9, but the manuscript lacks information about off-target effect assessment and verification of K2 absence at the mRNA level. Using scramble sgRNA alone is insufficient as control. Please specify if off-target effects were evaluated and by which methods.
Was the scramble sgRNA sequence checked against human and mouse genomes to confirm specificity?
- Cell culture details for EV production (seeding density, harvest density, cell viability) are missing. According to MISEV, these characteristics should be reported. Centrifugation acceleration and deceleration (brake) settings should also be included.
- There is no mention or imaging of EV morphology by electron microscopy (TEM, SEM, cryo-EM). Such data are essential to confirm the presence of sEVs, especially as the article’s conclusions depend on this EV subtype. EV morphology is currently best assessed for smaller EVs using high-resolution imaging techniques
- In the d-STORM section, the authors should add more methodological details as recommended by MISEV, including:
- EV labelling protocol details
- Coverslip preparation and coating protocols
- Fixation protocols and controls for affinity separation
- Microscope components, imaging settings (laser power, configuration)
- Multicolor imaging alignment and chromatic aberration corrections
- SMLM image processing, photophysical characterization, and analysis algorithms
- Authors published a paper in journal of Extracellular vesicles previously and the d-STROM paragraph was more detailed
- Antibody dilutions used for Western blots are missing and should be added.
Results :
- Again, electron microscopy images of EVs are required to confirm sEV morphology. How can the author claim that K2 does not affect EV morphology based solely on NTA results? Additional experiments, such as TEM imaging, are needed to confirm the morphology.
- The authors should verify whether their differential ultracentrifugation (dUC) protocol enriches for sEVs in the 100,000 x g pellet compared to other pellets (e.g., 10,000 x g) ? Was nanoparticle tracking analysis (NTA) conducted on these other fractions ? Figure 1A shows EV sizes > 200 nm; a shift in peak size should be quantified.
- A graph showing EV concentration (with statistical analysis) per sample should be added to verify that K2 knockout does not alter EV yield. Figure 1A suggests shifts in EV size and concentration between WT and K2-KO samples. It’s seen have a shift of EVs size (150 nm and 100 nm respectively) and concentration (peak around 1.0x10^9 particles/mL for 4T1 EVs and 1,60 x10^9 particles/ mL for 4T1-K2-KO EVs).
- Figure 1D, western blots are too small; band visibility (CANX, CD9, CD81 in 231 cells) is poor compared to 4T1 cells. Larger images or higher resolution blots are needed. The 4T1 total cell lysate sample is missing.
- According to MISEV, three EV markers should be assessed: at least two positive and one negative. CD9 and CD81 are both tetraspanins (transmembrane proteins); a cytosolic EV marker (e.g., Flotillin, Heat Shock Proteins) should be included.
- Confocal microscopy images do not convincingly show colocalization of K2 with nuclear markers (No yellow points). Without Z-stacks or 3D reconstructions, nuclear localization cannot be confirmed. In the current images, puncta can also be observed in the cytoplasm and in other cellular organelles. Proximity ligation assay (PLA) could help confirm nuclear presence. Additionally, no DAPI or Hoechst images are shown to visualize nuclei.
- Improve quality of Figures 2C and 2D: reposition quantification numbers below blot images for clarity, add graphs with standard deviation and statistical tests to support changes in α-SMA and FAP expression, and indicate molecular weights on blots. In the same experiments, check whether α-SMA and FAP mRNA levels increase via RT-qPCR to confirm possible nuclear effects of sEVs on recipient cells.
- In the tumorosphere invasion assay with fibroblast-conditioned media (CM) (Figure 2E-F), why compare sEV-educated fibroblast CM only to naïve fibroblast CM and not to CM from fibroblasts treated with K2-KO sEVs? Including this control would strengthen conclusions. The literature already reports that CAFs CM influences tumor invasion; this experiment should specifically dissect the role of K2-sEVs.
Discussion and Conclusion:
The conclusion section should be expanded, particularly to outline future research directions.
Final Recommendation: Major Revision
The study presents novel and potentially impactful data. However, greater methodological rigor and clarity, especially regarding EV characterization and functional controls, are needed to ensure the robustness and reproducibility of the findings.
I look forward to reviewing a revised version addressing the points above
Reviewer 3 Report
Comments and Suggestions for Authors
The manuscript by Yousafzai et al. with a title: “Role of Kindlin-2-Expressing Small Extracellular Vesicles in the Invasiveness of Triple Negative Breast Cancer Tumor Cells” provides mechanistic in vitro study using kindlin-2 knockout breast cancer cell lines and their extracellular vesicles (EVs). The experimental design is straightforward and includes all necessary controls. First, the authors characterize the isolated EVs using standard approaches such as detection of markers by Western Blot and size and concentration measurements using nanoparticle tracking analysis. The author then compare the effects of conditioned medium and EVs from parental and kindelin-2-knockout cells on breast cancer cell invasiveness in 3D invasion assays and fibroblast activation. An interesting finding of the report is that deletion of kindelin-2 in cell lines results in the release of kindelin-2 deficient EVs, which can then enter the cancer cells and fibroblasts and differentially activate cell signaling depending on presence or absence kindelin-2. The effects presented in 3D invasion assays are very strong,, suggesting that kindelin-2 could be a promising target protein for further studies and therapies. The manuscript is well written and contains all the necessary methodological details to allow readers to replicate the results.
Round 2
Reviewer 1 Report
Comments and Suggestions for Authors
The authors have greatly improved the quality of the manuscript and it is now fit for publication.
Reviewer 2 Report
Comments and Suggestions for Authors
The authors have responded to all of my comments with solid arguments, although some of the questions (particularly those regarding the controls in Figure 2E-F) were not fully understood. If the requested experiments cannot be performed for various reasons, the revisions made to the text help to mitigate these shortcomings